# Immunosenescence in Neurological Diseases—Is There Enough Evidence?

**DOI:** 10.3390/biomedicines10112864

**Published:** 2022-11-09

**Authors:** Paulus S Rommer, Gabriel Bsteh, Tobias Zrzavy, Romana Hoeftberger, Thomas Berger

**Affiliations:** 1Department of Neurology, Comprehensive Center for Clinical Neurosciences and Mental Health, Medical University of Vienna, 1090 Vienna, Austria; 2Department of Neurology, Division of Neuropathology and Neurochemistry, Comprohensive Center for Clinical Neurosciences and Mental Health, Medical University of Vienna, 1090 Vienna, Austria

**Keywords:** immunosenescence, aging immune system, neurological disorders, multiple sclerosis, stroke, Parkinson’s disease, dementia, Alzheimer’s disease

## Abstract

The aging of the immune system has recently attracted a lot of attention. Immune senescence describes changes that the immune system undergoes over time. The importance of immune senescence in neurological diseases is increasingly discussed. For this review, we considered studies that investigated cellular changes in the aging immune system and in neurological disease. Twenty-six studies were included in our analysis (for the following diseases: multiple sclerosis, stroke, Parkinson’s disease, and dementia). The studies differed considerably in terms of the patient groups included and the cell types studied. Evidence for immunosenescence in neurological diseases is currently very limited. Prospective studies in well-defined patient groups with appropriate control groups, as well as comprehensive methodology and reporting, are essential prerequisites to generate clear insights into immunosenescence in neurological diseases.

## 1. Introduction

The aging process in the human body is subject to several physiological processes that lead to increasing susceptibility to diseases such as malignancies and infections [1]. The importance of age in defending against infectious agents became evident during the COVID-19 pandemic, when age was the major risk factor for a severe or fatal outcome, along with various pre-existing conditions [2,3]. In this context, the concept of immunosenescence has received further scientific attention. The concept of immunosenescence was introduced by Walford [4] and refers to age-related changes in the immune system [5,6]. However, clear immunological changes or biomarkers defining the process are still lacking. Both the innate and adaptive immune systems change during the aging process, with the adaptive immune system currently receiving more attention [7,8]. Thymic involution plays a crucial role [9]. Thymic involution begins at birth, is likely to be evident from puberty onwards, and leads to a decreased ability of the generation of naïve T cells. The number of naïve T cells as well as clonal diversity decreases with age, and vice versa with the proportion of memory T cells, which increase [10,11]. Cellular changes lead to an altered secretion profile of cytokines and chemokines with increasing age [12]. In addition, humoral immunity in the adaptive immune system is also subject to age-related changes. The renewal of B cells in the bone marrow is reduced, while the number of B cells in the peripheral blood seems to be unchanged. As an example of immune changes, the detection of age-associated B cells (ABCs, CD19+CD21-CD11c+ T-bet+) could be shown in mice [13]. It is controversial to what extent these cells differ from “normal” B cells by their B-cell receptor repertoire. They are supposed to be mainly activated by Toll-like receptors (TLR), differ in the secretion of cytokines, and lead to an enhanced antigen presentation to T cells. They are likely to play a role in autoimmunity and viral clearance [14]. The aging is thought to be accelerated by chronic infections such as cytomegalovirus (CMV) or autoimmune diseases [9,15,16]. Chronic low-level inflammation is observed during the aging process [17,18]. Changes also occur in the natural killer (NK) cells compartment during the aging process. Although there is an increase in NK cells in older people, the immune surveillance decreases. However, the proportion of subpopulations of NK cell phenotypes varies between studies [19,20] Data on the innate immune system in general are scarce. Changes of the immune system are numerous and difficult to capture in their completeness, reflecting the heterogeneity and complexity of the aging immune system and strongly limiting the establishment of clear definitions or biomarkers. Recently, the role of the aging immune system has been discussed in the onset or progression of neurological disorders [10].

The aim of this review is to provide an overview of studies addressing the topic of immunosenescence and neurological disease, in order to summarize the status quo of evidence.

## 2. Methods

PubMed was searched for cellular studies on the terms “immunosenescence” AND “neurological disorders”, as of 27 September 2022. The search resulted in 228 articles. Of the 228 articles found, after excluding 114 review articles, 114 original articles were considered for inclusion in the analysis. After further excluding case reports or studies that did not primarily address neurologic disease or age-related changes or did not perform immunologic analyses, 26 studies of multiple sclerosis (MS), Parkinson’s disease, dementia, and stroke were included in this review.

### 2.1. Multiple Sclerosis

#### 2.1.1. Epidemiology and Pathogenesis

Multiple sclerosis is the most common neuroimmunological disease in young adulthood with more than 2.5 million patients affected worldwide. Women are affected 2.5 to 3.0 times as often as men, with the proportion of newly diagnosed women and men equalizing in later and progressive courses [21,22]. It is noteworthy that the pathophysiological condition differs between the courses. While focal inflammation is the main driver of disease activity in the relapsing forms of the disease, the neurodegenerative components predominate in later disease courses, although focal inflammation is still detectable [23]. Clinically relevant and possibly reflective of the process of immunosenescence is the observation, that after 40 years of age, differences in efficacy between the various immunomodulative or immunosuppressive therapies used in MS seem to be largely equalized [24], or are even discussed as discontinuable in stable courses [25,26].

#### 2.1.2. Studies on Immunosenescence in MS

Seven studies concerning MS are included in this review (see also Table 1). A recently published study examined CD4 T cell activation and the frequency of cytotoxic CD4 cells, with a decrease in CD4 cells in relation to CD8 cells often viewed as a correlate of immunosenescence. Forty untreated MS patients were compared with 49 non-MS patients (without any underlying inflammatory disease in controls and no prior immunomodulatory treatment in MS patients, most MS patients had mild disability (Expanded disability status scale, EDSS ≤ 2.5)). There were no differences between groups for the frequencies of most CD4 and CD8 subgroups. MS patients showed an age-related increased activation of CD4 cells and increased numbers of cytotoxic CD4 cells, which was particularly pronounced in those over 60 years of age. MS patients were found to lack the upregulation of CTLA-4 on CD4 and CD8 cells [27]. Eschborn et al. examined immunological signatures in blood and cerebral spinal fluid (CSF) from relapsing-remitting (RRMS) patients, primary progressive (PPMS) patients, and matched controls [28]. The participants were grouped into ‘older’ (>50 years) and ‘younger’ (≤50 years) individuals. The patients were not treated for at least three months with any disease-modifying therapy (DMT). In the blood (RRMS = 38, PPMS = 40, controls = 40), an age-dependent decrease in immunoregulatory molecules on memory CD8 cells (killer cell lectin-like receptor subfamily G member 1 (co-inhibitory protein: KLRG1), lymphocyte-activation gene 3 (inhibitory protein also referred to as CD223:LAG3), and cytotoxic T-lymphocyte-associated protein 4 (inhibitory protein receptor also referred to as CD152: CTLA-4) in controls was seen, which was not observed in MS patients. In CSF (RRMS = 51, PPMS = 36, controls = 85), PPMS patients showed an age-associated decrease in various immune compartments (B and T cells, plasma cells, and NK cells) in contrast to RRMS patients and controls. Both studies point in the direction of immune disbalance in elderly MS patients involving both CD4 and CD8 cells. The impact on elderly MS patients and MS patients in progressive course remains to be shown. Another study investigated whether patients with lipid-specific oligoclonal IgM bands (LSOCM+), a possible marker for aggressive MS [29], may have an altered immune profile during aging compared to those without LS-OCM. The cross-sectional study included 263 patients (72 had corresponding LSOCM, 191 did not). LSOCM− patients showed a significant decrease in mononuclear cells (including CD4, CD8, and B cells) with increasing age compared to LSOCM+ patients. Similarly, these patients were found to have increased CMV serum antibody titers. It is unclear how CMV and LSOCM interact together. In contrast, LSOCM+ patients showed an age-dependent increase in T cell immunoglobulin and mucin-domain containing-3 (transmembrane protein on various immune cells with inhibitory functions: TIM-3). An increase in programmed cell death protein 1 (inhibitory protein on various immune cells also referred as CD279: PD-L1) was observed in both groups. Disease duration had no effect, but these changes correlated with disability. It is not clear how CMV and LSOCM interact. To what extent these differences are associated with more aggressive courses or with CMV can only be speculated [30].

In one study, Hecker et al. examined telomere length (leukocyte telomere length: LTL), another marker whose decrease is a surrogate of increasing age and possibly immunosenescence, in 40 RRMS, 20 PPMS, and 60 controls [31]. About 75% of RRMS patients were treated with interferon-beta, whereas 60% of the PPMS patients received corticosteroids on an interval basis. The data were correlated with clinical courses after 10 years, and the LTL of a subset (10 patients) of the subjects could be plotted over time. RRMS patients exhibited a relative shortening of LTL compared with PPMS and control subjects. Shorter LTL was associated with earlier conversion to SPMS. In 122 patients initially treated with interferon beta, fingolimod, alemtuzumab, or natalizumab, the extent to which immunotherapies may have independent or additive effects on the aging immune system was investigated. To this end, the number of T cell receptor excision circuits (TRECs) and κ (kappa)deleting recombination excision circuits (KRECs) in T- and B-lymphocytes, as markers for T- and B-cell development [32], was examined.

The samples were analyzed at baseline and after 6 and 12 months of treatment. In therapy-naive patients at baseline, a significant correlation was found between decreased levels of TRECs and KRECs and increasing age. The therapies showed no clear influence on the number of TRECs and KRECs [33]. In a study of 32 MS patients, 60 rheumatoid arthritis (RA), and 40 control patients, TRECs were found to be decreased in RA and MS compared with healthy peers. Sub-cohorts of RA and MS patients had significantly increased numbers of CD4+CD28 cells [34]. The same study group investigated TRECs and CD4+CD28− as markers of immunosenescence in 175 patients with autoimmune diseases (MS = 70) compared with controls (*n* = 60). TRECs were significantly decreased in RA and MS. Similarly, CD4+CD28− were present at a higher percentage compared to controls, and a correlation with CMV infection and HLA-DR4 was demonstrated for RA (not studied for MS) [35].

**Table 1 biomedicines-10-02864-t001:** Immunosensenescene and MS.

MS	Involved Patient Groups	Numbers, Age	Material	Assay Readouts	Main Results
**Zuroff (2022) [27]**	MS controls	40, 39 (35–55) 49, 49 (34–65)	peripheral blood	T and B cells naïıve (CCR7+CD45RA+), CM (CCR7+CD45RA), EM (CCR7-CD45RA), TEMRA (CCR7-CD45RA+)	altered age-associated activation of CD4 T cells and reduction of CD4 RTEs in MS controls exhibited increase of co-inhibitory CTLA-4 CD8 and CD4 T cells
**Eschborn (2021) [28]**	A: RRMS PPMS HC B: RRMS PPMS non-inflammatory controls	A: 38 40 40 51 36 85 grouped in ‘older’ (>50 years) and ‘younger’ (≤50 years)	peripheral blood CSF	Lymphocytes B cells, CD4 T cells, CD8 T cells, CD8 naïve, CD8 memory, CD8 EM, CD8 CM	A: An age-dependent decrease in the expression of immunoinhibitory molecules (KLRG1, LAG3, CTLA-4 on memory CD8 T cells) was abrogated in MS T cells of old patients with MS displayed increased intracellular expression IFN-γ and TNF-α on stimulation Controls exhibited a strong age-dependent increase in costimulatory molecule CD226 on memory and EM CD8 T cells B: An age-dependent decrease in counts of lymphocytes, B and T cells and NK cells in patients with PPMS
**Picon (2021) [30]**	MS grouped in M− (LS-OCM negative) M+ (LS-OCM positive)	263 (16–65) 191 (16–65) 72 (18–62) grouped in >45 years and ≤45 years	paired peripheral blood and CSF	main subset of CSF (CD4+ and CD8+ T cells, CD19+ B cells, NK cells, monocytes and soluble factors (PD-L1, TIM-3, NfL)	M− exhibit an age-related decrease of mononuclear cells, NK cells, CD4, CD8 and B cells compared to M+. M− exhibited an age-related decrease of CD4 and CD8 T cells producing INF-γ, TNF-α and GM-CSF M− >45 years had a lower number of CD4 and CD8 T cells producing INF-γ, TNF-α and GM-CSF
**Hecker (2021) [31]**	RRMS PPMS controls	40, 50 (24–67) 20, 47 (26–68) 60, 51 (18–74)	peripheral blood	LTL from leukocytes	LTL was shorter for RRMS than for PPMS and HC An age-related negative association for LTL for all groups was shown shorter LTL in RRMS was associated with conversion to SPMS
**Paghera (2020) [33]**	RRMS controls	122, four groups (17–60 years) 235 age-matched	peripheral blood	TRECs KRECs	age-dependend decrease of TRECS in HC and MS KRECS remained stable, but decreased in an age-related manner
**Thewissen (2007) [35]**	controls autoimmune MS	60, 44 (20–85) 175 70, 44 (16–69)	peripheral blood	TRECS CD4+CD28^null^	An age-inappropriated low number of TREC in MS (and RA) compared to controls was shown. A non-significant decline in TRECs for disease duration in MS CD4+CD28^null^ was shown as well as an increased frequency of these cells in MS
**Thewissen (2005) [34]**	controls RA MS	40, 45 (21–75) 60, 56 (34–84) 32, 39 (16–58) patients were grouped in three aging groups (15–40, 41–60 and >61 years)	peripheral blood	CD4+CD28^null^	TREC in MS were lower than in controls (not statistically significant) frequencies of CD4+CD28^null^ did not differ at young age between MS and controls, but frequency increased with aging for MS compared to controls

Abbr.: CM = central memory, EM = effector memory, KRECs = K-deleting recombination excision circles, LTL = leukocyte telomere length, MS = multiple sclerosis, RRMS = relapsing-remitting MS, PPMS = primary progressive MS, SPMS = secondary progressive MS, RA = rheumatoid arthritis, RTE = recent thymic emigrants, TRECs = T-cell receptor excision circles.

#### 2.1.3. Conclusions

The few studies that have examined age-related changes in the immune system in MS have focused primarily on T cells, thymic emigrants, and LTL. The extent to which older MS patients have an abnormally increased frequency of CD4 T cells with activated and cytotoxic effector profiles [27] cannot yet be conclusively answered. A disbalance between costimulatory and immunoregulatory signaling by CD4 and CD8 T cells has been demonstrated, favoring a proinflammatory phenotype [27,28]. A pattern of premature immune aging has been demonstrated in the CD8 T cell compartment as well as in telomere length in leukocytes [28,31]. In MS patients, a moderate induction of T cell tolerance and activation of innate immunity with growing age seem to occur [30]. Both genetic factors, such as HLA DR4, and environmental factors, such as CMV infection, could accelerate immunosenescence in MS [27,35]. The extent to which these changes are of pathophysiological significance in aging MS patients is not known. The evidence is too limited to draw general conclusions about the impact of immunosenescence in MS.

### 2.2. Stroke, Cerebrovascular Disease

#### 2.2.1. Epidemiology and Pathogenesis

Cerebrovascular disease, and especially acute ischemic stroke, is a major cause for long-term morbidity, disability, and mortality [36]. Ischemic stroke leads to a focal neurological deficit based on a local injury due to a vascular cause. Within a time-window and with respective vascular findings, different therapeutic options such as intravenous thrombolysis and endovascular thrombectomy can be applied. In the further course, the optimization of risk factors and appropriate anticoagulant therapy should prevent a further event [37,38]. With increasing life expectancy and a consecutively aging society in most parts of the world, the prevalence of stroke is increasing. After the acute event, the regeneration potential of the patient is decisive for the outcome [39]. In this context, the immune system is of great importance, which not only fights off foreign patterns, but is also involved in the clearance and repair of debris and activated by damage-associated molecular patterns (DAMPS) [40,41,42]. Only one study was included in our review investigating immunosenescence after stroke (see Table 2).

#### 2.2.2. Studies on Immunosenescence in Stroke and Cerebrovascular Disease

This study investigated the immune profiles (leukocyte gene expression) by transcriptomics. Peripheral blood RNA from two cohorts (‘stroke’ and ‘validation’ cohort) with acute ischemic stroke was measured using whole-genome microarrays, and transcriptomic changes associated with increasing age were determined. The changes in expression profile were compared with age-associated genetic expression profiles from non-stroke studies (*n* = 3973). Verified genes (detected in both cohorts) were analyzed. There was a decrease in receptors CR2, CD27, CCR7, and 5′-nucleotidase ecto (5′-NT) (enzyme serving as lymphocyte differentiation marker on immune cells, also referred to as CD73: NT5E). The detected genes were implicated in altering B-cell receptor signaling, lymphocyte proliferation, and leukocyte homeostasis. Forty-three of the 69 genes were also verified in published non-stroke studies [43].

#### 2.2.3. Conclusions

Only one study, using transcriptomics, examined the immune changes occurring after stroke and related these to those of the aging process. However, the evidence is far too limited to draw any conclusions here.

### 2.3. Neurodegenerative Disorders

In addition to neuroinflammatory diseases, where the influence of the immune system is clear but the evidence of immunosenescence is very limited, the importance of the immune system in the development/progression of neurodegenerative diseases such as Parkinson’s disease (PD) or dementia has been discussed.

#### 2.3.1. Parkinson’s Disease

##### Epidemiology and Pathogenesis

Parkinson’s disease (PD) is a neurodegenerative disorder of the extrapyramidal system with the core symptoms of rigidity, tremor, and akinesia. PD is the second most common neurodegenerative disease after Alzheimer’s disease (AD). It is primarily a disease of the elderly, and its prevalence peaks between 85 and 89 years with a predominance of males (male–female ratio up to 4:1) [44]. Patients with disease onset before the age of 50 are rare. Besides idiopathic PD, there are hereditary variants and “secondary” forms resulting from medication, cerebrovascular disorders, trauma, or toxicity. The cause of the symptoms is a dopamine deficiency, which is due to the destruction of neurons in the substantia nigra. The imbalance of neurotransmitters in the basal ganglia explains the various symptoms [45]. In recent years, inflammatory pathways have been discussed in the pathogenesis of PD [46]. Interestingly, anti-inflammatory drugs may play a protective role [47].

##### Studies on Immunosenescence in Parkinson’s Disease

Studies on immunosenescence in PD are rare. Four studies were included in the review (Table 3). One study examined the extent to which CMV seropositivity affects the T- and NK-cells of 31 patients with mild to moderate PD, 33 age-matched controls, and 30 young controls. All PD patients were CMV-positive (+), as opposed to between 73 and 76% of peers and young controls, respectively. The proportion of both, effector memory T cells re-expresses CD45RA (TEMRA) cells, and CD57+CD56+ T (NKT) cells were significantly reduced in PD patients compared with their CMV+ peers. The frequency of NKT cells did not differ from young CMV+ [48]. A study of 61 newly diagnosed PD patients and 63 age-matched control subjects investigated whether T cell dysregulation may contribute to the development of PD. Telomere length as well as markers of cell aging (p16INK4a and p21CIP1/Waf1) were examined. CD8-TEMRA cells [49] in PD were reduced compared to controls, as was aging marker p16INK4a. Latent CMV infection (CMV+) in the controls resulted in a significant increase in the numbers of CD8+CD28lowCD57hi cells in the controls, but no significant difference for CMV+ and CMV− PD patients [50]. Striking was the difference in the CMV positivity rate of PD patients (48%) compared to the previously mentioned study with a 100% positive rate [48]. Vida et al. studied the effects of oxidative stress and functional capacity of immune cells in newly diagnosed PD patients. Blood from PD patients was analyzed for neutrophil and mononuclear cell functional capacity (see Table 3) at various time points during the disease compared to controls. Lymphoproliferative capacity was decreased in PD patients compared to controls. Higher levels of oxidative stress were found in PD patients (lower glutathione peroxidase activity and higher levels of oxidized glutathione and malondialdehyde). PD medication had no effects on oxidative stress markers [51]. In another study, blood samples from 41 early PD patients (mean disease duration: 4.3 years) and 41 age-matched controls were analyzed by immunophenotyping for T cell activation (HLA-DR, CD38) and senescence (CD28, CD31, CD57). The CD8 markers of senescence were significantly reduced in PD (reduced CD57 expression, “late differentiated” CD57loCD28hi cells, and “TEMRA” cells). In contrast, CD28 as an activity marker was increased in PD. A possible influence of CMV infection was excluded [52].

##### Conclusions

Evidence for an altered immune system in PD is very limited. The four studies mainly examined T cell subpopulations. The peripheral immune profile in PD differed from a “normal” elderly population, and a reduced immunosenescence of T cells in PD patients is suggested [48,50,51,52]. Parts of the innate immune system were investigated in only one study [51]. The humoral immune response was not studied in detail. The extent to which “abnormal” immune aging may contribute to the development of PD or vice versa cannot be answered based on the studies.

#### 2.3.2. Dementia and Alzheimer’s Disease

##### Epidemiology and Pathogenesis

More than 55 million people worldwide currently suffer from dementia. Every year, a further 10 million patients are added to this number. Around 60–70% of all dementia patients suffer from AD. The numbers are expected to rise to 78 million in 2030 and around 139 million in 2050 [53]. The incidence of AD has been increasing in Europe in recent years and is approximately twice as high for women than for men [54]. Amyloid beta deposits, neurofibrillary tangles and plaques are the main characteristics of the disease. There is no cure for AD, and the disease cannot be stopped or reversed, albeit potentially disease-modifying drugs are emerging (e.g., lecanemab). Sometimes, a short-term improvement of symptoms can be achieved. Current treatment consists of symptomatic treatment, and acetylcholinesterase inhibitors or NMDA receptor antagonists aim to slow the progression of the disease [55].

Several studies have investigated the role of the aging immune system in dementia. Fourteen studies were included in our review (see Table 4).

##### Studies on Immunosenescence in Dementia and Alzheimer’s Disease

In the retrospective cross-sectional Mugello study (enrolling people aged 90 and older; about 65% of all residents over 90 are covered by the Mugello study; data were gathered in 2009), inflammatory markers in peripheral blood and the development of dementia were investigated in 411 (110 male and 301 female) participants. The development of dementia was reported for the enrolled participants in 73 cases (17.8%). Dementia patients were older, had suffered stroke more frequently, and had higher peripheral blood lymphocyte counts and a higher lymphocyte-to-monocyte ratio. These results suggest a possible influence of the immune system, but further phenotyping on immune cells was not available [56]. One of the first studies on immunosenescence compared immunological functions in AD with elderly and young controls. AD patients and the elderly showed no significant differences in lymphocyte surface markers. Both groups showed decreased polyclonal B-cell reactivity compared to young controls. In AD patients, the activity was mainly decreased. A deficit in leukocyte inhibitory factor (LIF) release was found in AD patients. While a decrease in polymorphonuclear cell (PMN)-mediated functions and phagocytosis capacity of monocytes was observed in elderly subjects, only a decrease in PMN response was observed in AD (for more details see Table 4) [57]. The extent to which oxidative stress may affect T cell activation and development in AD was investigated, using the 3-nitrotyrosine (3-NT) proteome of T cells derived from the blood of probable or possible AD patients and control subjects without dementia. Using proteomics, ten proteins with elevated 3-NT levels were identified in AD patients. These proteins are mainly involved in energy metabolism, cytoskeletal structure, intracellular signaling, protein folding and turnover, and antioxidant response [58]. Another study investigated whether AD patients have a CMV-specific immune profile compared to their peers. Blood samples from 50 AD patients and 50 [52] age-matched controls were analyzed for HLA type, CMV serostatus, and further phenotype markers (CMV-specific CD8 immunity, and then further classified with CD27, CD28, CD45RA, and CCR7). Mean Mini-Mental-State-Examination (MMSE) was 19.9 ranging from 10 to 27. CMV seropositivity was found in 84% patients with AD and for 78% of controls. AD patients with CMV+ had a significantly lower percentage of CMV-specific CD8 T cells than the control group (*p* = 0.0057). In general, CMV+ probands have a lower proportion of naïve CD8 cells and a higher proportion of effector CD8 cells compared to negative ones. Although CMV+AD patients had fewer CMV-specific CD8 cells than non-AD probands, there was no difference in CD8 subgroups [59].

The production of amyloid-β42-peptide is an important step in the pathogenesis of AD. The extent to which amyloid-β42 peptide subsequently triggers an immune response is the subject of scientific debate. In one study, the number of CD19 B cells and B-cell subpopulations were examined in patients with moderate–severe as well as with mild AD compared with healthy controls. There was a significant decrease in naive B cells (IgD+CD27-) and an increase in double-negative (DN, IgD-CD27-) memory B lymphocytes. Both the amount and the chemokine profile of B cells were related to the severity of AD [60]. The importance of DN B cells was supported by another study, where an increase in the proportion of the IgD(-)CD27(-) memory B-cell (double negative, DN) population in the elderly was reported [61]. In another study, DN B cells were further characterized. Enrolled AD patients were moderately to severely affected (MMSE ≤ 17). The expression of the inhibitory receptors CD307d and CD22 on these cells from young and old people was investigated. The ability to activate DN-B cells by the simultaneous use of innate (CpG) and adaptive (α-Ig/CD40) ligands was investigated. The stimulation and activation of DN-B cells was successful when they bound both BCR and TLR9, but in contrast to controls this did not occur by a single receptor, representing an impaired activation potential.

Reactivation of the enzyme telomerase was reduced in AD patients [62]. Peripheral blood mononuclear cells (T, B, and NK cells) (PBMC) were immunophenotyped in 51 AD patients (29 with mild dementia and 22 with moderate dementia) and 51 peers. Additional surface markers (CD25, CD28, CD57, CD71, CD45RA, and CD45RO markers on CD4+ and CD8+ cells) were analyzed. IL-2, IFN-γ, IL-10, and TNF-α were examined after stimulation with beta-amyloid (betaA) fragments in subgroups (AD *n* = 30, controls *n* = 20). AD patients had significantly fewer circulating B and CD8+CD28+ cells and a high number of CD8+CD71+CD28+ cells. Significantly decreased IL-10 production was observed after stimulation with betaA, but this did not correlate with dementia severity [63]. Schindowski et al. studied the distribution and apoptosis of lymphocyte subsets of peripheral blood mononuclear cells (PBMC) in AD and peers in T- (CD4+ CD3+, CD8+ CD3+), B- (CD19+), and NK cells (CD16++CD56+). Aging in general was associated with a higher rate of apoptosis. However, in AD patients (mean MMSE: 18.8 +/− 11.2), there was increased apoptosis of CD4+ T and NK cells. B-cell lymphoma-2 (anti-apopoptic enzyme: Bcl2) levels in T cells were significantly increased in mild AD [64]. The subsets of T, B, and NK cells were determined in 43 patients with AD (mean MMSE 17.9, ranging from 11 to 22) and 34 age-matched control subjects by flow cytometry. In AD patients, there was a significant decrease in CD3+ and CD19+ lymphocytes. For CD3+ cells, there was an increase in the CD8+ subpopulation but an increase in CD4+ T cells. The CD4+/CD8+ ratio did not change significantly. There was no difference in NK cells between the groups [65]. Pro-inflammatory cytokines (IL-1ß, IL-2, IL-6, and TNF-α) and the soluble receptors sIL-2R, sIL-6R, and sTNF-αR were measured in the CSF and serum of 20 AD patients and 21 control subjects. Thereby, the levels were either undetectable (IL-1ß, IL-2, TNF-α) or significantly decreased in AD patients. In addition, mitogen-stimulated blood cultures from 27 AD and 25 control subjects showed that pro-inflammatory cytokines (IL-6, Il-12, IFN-γ, and TNF-α) and anti-inflammatory cytokines (IL-5 and IL-13) were significantly reduced in AD patients [66]. Busse et al. investigated the innate immune system with regard to the pathogenic role in AD. The number of CD14+ monocytes and the frequency of HLA-DR, CD80, and CD86 expression were examined in controls (aged 20–79 years) and AD patients at the time of diagnosis and repeatedly after the initiation of rivastigmine treatment. The numbers of CD14+ monocytes were constant in the AD group over time and did not differ significantly from controls. The expression of HLA-DR, CD80, and CD86 on monocytes increased with the age of enrolled subjects. There were no differences between controls and AD patients over time [67]. Post-mortem, 112 brains of patients with post-stroke dementia, vascular dementia, mixed dementia, and Alzheimer’s dementia were examined for their immune profile. These were compared with non-stroke dementia subjects and age-matched controls. Five brain regions were analyzed for their cytokines and chemokines by multiplex array. Of the 37 analyses, 16 analytes were assessed. There were broad variations of C-reactive protein (CRP) and interleukins (INF-α and tumor necrosis factor) in the low range between the groups. Decreased levels of interleukins were found in patients with dementia compared with those without (IL-1ß, IL-6, IL-7, IL-8, IL-16). IL-6 and IL-8 were lower in all regions studied in post-stroke dementia compared to patients without stroke or dementia [68]. Transcriptomics was used to study immunological markers in the brains of patients with probable dementia with Lewy bodies (DLB) post-mortem. Small extracellular vesicles (SEVs)—enabling RNA transport between the brain and peripheral circulation—were selected for analysis. The SEV RNA profiles of 10 DLB patients and 10 control brains without dementia were analyzed by RNA sequencing. Significant decreases in proinflammatory genes (IL1B, CXCL8, and IKB) were detected in the brains of DLB patients compared with control subjects. Altered network analyses emphasize the influence of the immune system, but also the importance of dysfunction of the ubiquitin–proteasome system, and DNA repair in DLB pathology [69]. The same group examined the transcriptomics of Lewy body dementia (LBD) brains after death in the anterior cingulate and dorsolateral prefrontal cortex using next-generation RNA sequencing compared to other forms of dementia (DLB and patients with Parkinson’s Disease with Dementia [PDD]) and the brains of persons without dementia. Twelve genes were found to be significantly altered (MPO, SELE, CTSG, ALPI, ABCA13, GALNT6, SST, RBM3, CSF3, SLC4A1, OXTR, and RAB44) in LBD compared with controls without dementia. Several proinflammatory cytokine genes were downregulated, as well as mitochondrial dysfunction and increased oxidative stress [70].

**Table 4 biomedicines-10-02864-t004:** Immunosenescence and dementia and Alzheimer’s disease.

Dementia	Involved Patient Groups	Numbers, Age	Material	Assay Readouts	Main Results
**Lombardi (2021) [55]**	Demented controls, non-demented	73, 94.06 +/− 3.66 338, 92.81 +/− 3.14	peripheral blood	WBC	increased lymphocyte count in demented higher lymphocyte-monocyte-ratio (LMR) in demented patients only demented (without history of stroke) had higher lymphocyte and lower monocyte count, higher LMR
**Rajkumar (2021) [69]**	probable DLB (dementia with lewy-bodies) controls, non demented	10, age not reported 10, age-matched according the authors	peripheral blood RNA-seg	SEVs DEG (differentially expressed genes)	Downregulation in DLB: - RNA expression levels of pro-inflammatory genes (IL1B, CXCL8, and IKBKB) - RNA expression levels of proapoptotic genes (BID and TNFRSF1A, ubiquitin proteasome system (UPS) associated UBE3A, USP47, and PSMD4) Upregulation in DLB: - PTPRF, MIR556, and SMG9
**Rajkumar (2020) [70]**	LBD (lewy body dementia) DLB (dementia with lewy bodies) PDD (PD dementia) controls, non-demented	14, age not reported 7 age did not differ, between the groups 7 7	post-mortem brain tissue RNA-seq	DEGs	downreguation in LBD: MPO, SELE, CTSG, ALPI, ABCA13, SST, RBM3, CSF3, SLC4A1, OXTR, and RAB44 upregulated in LBD: GALNT6
**Tramutola (2018) [66]**	AD controls, non demented	19, 76.94 +/− 9.44 19, 71.20 +/− 6.88	peripheral blood	proteomics 3-NT proteome in CD3+	upregulation of 10 proteins: PIK3R2, ANXA2P2, HSPA8, INPP4B, TADA2B, DPYSL2, ANXA11, CAT, ATP5A1 effects of T-cell on energy metabolism, cytoskeletal structure, intracellular signaling, protein folding and turnover, and antioxidant response
**Chen (2016) [68]**	AD VaD (vascular dementia9 mixed, dementia PSD, post stroke dementia) PSND, post stroke non.demented Controls	16, 83.9 +/− 1.9 17, 83.9 +/− 1.6 18, 84.5 +/− 1.2 20, 87.3 +/− 1.3 21, 85.0 +/− 1.0 20, 79.2 +/− 3.3	post-mortem brain tissue from 5 regions (FGM, TGM, FWM, TWM, HIPP)	multiplex analyte assays with 6 panels (pro-inflammatory, cytokines, chemokines, angiogenesis, vascular and bFGF)	lower concentrations of cytokines on general in demented patients compared to controls (higher concentrations of bFGF, ICAM-1, VEGF-C, VEGF-D) PSD vs. PSND had lower concentrations of IL-6,-8 but higher concentrations of IL-1α
**Busse (2015) [60]**	AD controls	23, age not reported 37, 20–79 years	peripheral blood for AD after initiation of rivastigmine treatment blood samples were collected: 12, 30 and 52 weeks	CD14, HLA-DR, CD80, and CD86	at time of diagnosis CD14+, HLA-DR expression, CD80/86 did not differ for AD vs. aged-matched controls after initiation of rivastigmine treatment no differences were detected over course of time
**Bulati (2014) [62]**	AD, moderate-severe AD, mild controls	20, 65–85 years 15, 69–91 years 15, 65–81 years	peripheral blood	CCR6, CCR7, CXCR3, CXCR4, CXCR5, CD19, CD27, IgD	Total and naïve B cells (IgD+CD27-) are increased in moderate-severe AD compared to controls Double negative B cells (IgD-CD27-) were increased in moderate-severe AD CCR6 was highly expressed on both AD groups CCR7 was highly expressed on total B cells in moderate-severe AD
**Martorana (2014) [64]**	AD, moderate-severe controls, young controls, eldery controls, age-matched controls, descendants of centenarians	8, 69–76 years 20, 25–40 years 20, 78–90 years 8, 69–76 8, 60–70	peripheral blood	IgD, CD19, CD22, CD27, CD307 cell proliferation: Ki67 telomerase activity (RTA)	AD patients had lowest RTA levels, no further significant differences from peers were detected
**Westman (2013) [61]**	AD controls, non demented	50, 77.5 +/− 6.9 50, 74.0 +/− 8.0	peripheral blood	CD3, CD4, CD8, CD19, CD27, CD28, CD45RA, CCR7 CMV seropositivity	proportion of CMV-specific CD8+ cells was significantly lower in AD than ND no difference for CD8 subpopulations
**Speciale (2007) [65]**	AD AD, mild AD, moderate severe controls, non demented	51, 72.2 (54–85 years) 29, 73.93 (54–85 years) 22, 69.90 (55–82 years) 51, 69.10 (54–87 years)	peripheral blood	CD3, CD4, CD8, HLA-DR, CD16, CD45RA, CD45RO, LFA1, CD25, CD28, CD71, CD57, cytokines (IL-2, IFN-γ, IL-10 and TNF-α)	CD8+CD71+ cells higher in AD CD8+CD28+ cells higher, and CD8+CD28− cells decreased in AD significant decrease in IL-10 after stimulation with Aß-protein in AD
**Richartz-Salzburger (2007) [58]**	AD controls, non demented	43, 70.9 +/− 8.2 34, 67.5 +/− 7.3	peripheral blood	CD3, CD4, CD8, CD19, CD16, CD56	AD patients showed a decrease in CD3+ (increased CD4+, but decreased CD8+), not altered CD4/CD8 ratio in AD patients CD19 cells were diminished
**Schindowski (2006) [57]**	AD controls, non demented	34, 73.4 +/− 3.5 34, 71.5 +/− 4.6	peripheral blood	CD4, CD8, CD16, CD19, CD56 apoptosis measuring Bcl2 staining	significant increase in the basal apoptotic levels in AD compared to controls increase in Thelper cells and decrease of cytotoxic/suppressor cells in AD
**Richartz (2005) [59]**	AD Serum AD CSF controls, non demented serum controls, non demented CSF	27, 70 (63–84) 20, 72 (62–88) 23, 68 (59–77) 21, 68 (59–82)	peripheral blood and CSF	cytokines (IL-1ß, IL-2, sIL-2r, IL-6, sIL-6r, TNF-α, TNF-αr)	reduced levels of all cytokines (IL-2 not detectable in CSF, and IL-1ß, IL-2 and TNF-α not detectable in serum), in CSF and serum for AD patients and reduced levels for all cytokines after mitogen-induced stimulation in AD
**Antonaci (1990) [56]**	Demented controls, age-matched controls, young	12, 70 (68–92) 12, 73 (66–83) 12, 26 (21–35)	peripheral blood	CD3, CD4, CD8 B Cell Polyclonal Response Antigen-Specific Induction System chemotaxis phagocytosis LIF production (LK assay), LDCF release	In AD LDCF was not decreased compart to aged controls, but compared to young controls PMN chemotactic responsiveness, phagocytosis, and killing were significantly reduced in AD, but not different to aged controls

Abbr.: AD = Alzerheimer’s Disease, Aß-protein = Amyloid-Beta-protein, FGM = frontal gray matter, FWM = frontal white matter, HIPPO = hippocampus, LDCF = lymphocyte-derived chemotactic factor, SEV = small extracellular.vesicles, TGM = temporal gray matter, TWM = temporal whiter matter, WBC = white blood cell count.

##### Conclusions

Compared to other diseases, more studies have been performed on immunosenescence and dementia. In addition to AD, DLB, and LBD, PDD as well as vascular and post-stroke dementias have been studied [68,69,70]. Peripheral blood as well as CSF [66] and brain tissue [68,70] have been investigated in these studies. Notwithstanding some conflicting data, most studies suggest that the inflammatory response is decreased in dementia compared with healthy individuals [68]. DLB patients have been shown to have a decreased expression of inflammatory mediators [69]; similarly, microglial activation does not appear to be pronounced in LBD, in contrast to AD [70]. Decreased numbers of B and T cells and altered chemokine and activation profiles have also been demonstrated in AD patients [57,60,62,65,66]. The role of oxidative stress in T cells appears to drive the aging process of immune cells in AD [58]. Overall, the picture is not uniform, e.g., reduced suppressor cell function with decreased IL-10 production suggests increased proinflammatory functions [63]. Due to the heterogeneity of the diseases and patients included as well as the materials investigated, the overall body of evidence precludes drawing a clear conclusion from the reported data.

## 3. Summary

Interest in the immune system in neurodegenerative diseases has been spurred by studies in recent years [71,72]. Although there are many articles on immunosenescence and neurological diseases, and although possible therapies to influence immunosenescence are discussed [73], the evidence is extremely scarce and partially contradictory. Twenty-six studies were included in our analysis. Compiled from these studies on immunosenescence, a total of 752 MS patients, 178 PD patients, 405 dementia patients, and 173 stroke patients have been studied so far. The respective cohorts ranged from 8 to 263 patients, reflecting the heterogeneity of the study situation. The pathogenesis of all the diseases studied involves the brain. It is not clear to what extent processes in the brain can be reflected by immunological markers in serum and blood, particularly bearing in mind the presence of the blood–brain barrier (BBB). Histopathological or CSF studies would thus be all the more important. Only a few studies included CSF analyses. The histopathological studies include 99 brains and are mainly from dementia research. The immune system beyond the BBB seems to be important in neurodegenerative diseases. Some studies in MS and AD [74,75,76] have analyzed immunological markers, but without illuminating the immune system in view of immunosenescence. Microglia are much more likely to be of crucial importance here [77]. Understanding the interplay between CNS-bound immunity and peripheral immunity is thus the foremost task [71]. Further studies are urgently needed to shed more light on immunosenescence in neuroinflammatory and neurodegenerative disorders. Prospective studies in clearly defined patient groups with appropriate control groups as well both comprehensive methodology and reporting are the imperative prerequisite to that end. Moreover, elderly patients are excluded from the approval studies in many diseases (e.g., multiple sclerosis), but it is precisely these patients who should be included due to possible side effects. In a German analysis, it was shown that 11.8% of the MS patients treated could not participate in the pivotal trials because of their age [78]. Including these patients is of the utmost importance. Our understanding of immunosenescence is currently increasing and suggests unexpected possibilities. It is known from animal experiments that the number of neural stem cells (NSCs) in the aging brain of mice decreases, and consequently, negative effects on the regeneration of a damaged brain are discussed [79]. In old mice, positive effects have already been shown through blood transfer from young mice [80]. However, the available evidence is too weak for meaningful conclusions and far from specifically related yet discussed therapeutic interventions [81].

## Figures and Tables

**Table 2 biomedicines-10-02864-t002:** Immunosenescence and stroke and cerebrovascular disease.

Stroke	Involved Patients Groups	Numbers, Age	Material	Assay Readouts	Main Results
**Sykes (2021) [42]**	Cohort 1: Stroke Cohort 2: Validation	94, 65.9 (38–90 years) 79, 63.8 (36–91 years)	peripheral blood (median 46 h after onset in cohort 1 and 35 h in cohort 2)	transcript of gene expression	69 genes associated with age from Cohort 1 that were confirmed in Cohort 2 (including CR2, CD27, CCR7, and NT5E) Functional analysis identified B cell receptor signaling and development, T helper cell differentiation, and IL-7 Association between age and post-stroke leukocyte gene expression was shown

Abbr.: NT5E = 5′-nucleotidase ecto.

**Table 3 biomedicines-10-02864-t003:** Immunosenescence and PD.

PD	Involved Patient Groups	Numbers, Age	Material	Assay Readouts	Main Results
**Valivova (2021) [47]**	PD controls, age-matched controls, young	31, 59 +/− 11.6 (Höhn & Yahr ≤ 2.0) 2.7 +/− 0.6 33, 56 +/− 11 20, 25 +/− 2.5	peripheral blood	CD3, CD4, CD8, CD56, CD57 CMV-seropositivity	PD: 100% CMV+ controls: 76% CMV+ T_EMRA_ cells were reduced in PD compared to CMV+ age-matched controls CD57+ cells were decreased in PD compared to age-matched controls, but increased compared to CMV+ young controls
**Kouli (2021) [49]**	PD controls	61, 67,4 +/− 7.1 (Höhn & Yahr ≤ 2.0) 63, 67.5 +/− 7.2	peripheral blood	T cells, RTE, Telomere length, gene expression marker hTERT, p16, p21	PD:48% CMV+ controls: 48% CMV+ reduction in lymphocytes and cytotoxic CD8+ T cells and T_EMRA_ in PD Association between CMV+ and CD8+CD57+ cells and CD8+T_EMRA_, expression of p16 was reduced in PD no difference in telomere length and hTERT between PD and controls
**Vida (2019) [50]**	PD elderly controls adult controls	45, 67 +/− 12 (Höhn & Yahr ≤ 2) 34, 74 +/− 11 20, 40 +/− 8	peripheral blood	neutrophils, lymphocytes adherence, chemotaxis, phagocytosis, NK cytotoxicity, lymphoproliferation, gluatathione peroxidase/reductase activity, lipid peroxidation	PD patients showed impairment of the adaptive immune functions (with lower lymphoproliferation) but not in the innate response PD had lower gluthatione reductase, but higher peroxidase activity
**William-Gray (2018) [51]**	PD controls	41, 68.4 +/− 6.3 (Höhn & Yahr ≤ 2) 41, 68.1 +/− 5.6	peripheral blood	CD3, CD4, CD8, HLA-DR, CD38, CD28, CCR7, CD45RA, CD57	Lower total lymphocyte counts in PD, but no difference in CD4/CD8 ratio reduced proportion of CD28loCD57hiCD8+ T cells, CD8+ TEMRA, but higher CD8+CM cells in PD CD57 was decreased and CD28 increased in PD

Abbr.: PD = Parkinson Disease, CMV = cytomegalovirus, RTE = recent thymic emigrant.

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
