# Peer review of "Immunosenescence in Neurological Diseases—Is There Enough Evidence?"

_biomedicines, 2022, doi:10.3390/biomedicines10112864_

Round 1
Reviewer 1 Report
The authors review the current state of evidence for the role of immunosenescence in neurological diseases, with a focus on multiple sclerosis, ischaemic stroke, Parkinson's disease and dementia. The topic is highly significant, with critical diagnostic and therapeutic implications. Often, the role of immunosenescence in neurodegenerative diseases is taken for granted, without much critical deliberation. The authors do a good job showing how limited is the scope of evidence specifically on this connection, and how often incompatible or conflicting results are present.
Of course, the subject also has critical therapeutic implications, especially with reference to geriatric patients whose specific immune response has been little studied. Often anti-inflammatory, immuno-stimulating or immuno-modulating therapeutic options for neurodegenerative diseases are considered uncritically, especially for the geriatric population. And here too the authors do a good job showing how limited the evidence base is for the effectiveness of such interventions. As the authors note considering one of the diseases “Interestingly, anti-inflammatory drugs may play a protective role.” But of course, that formulation also means that they also may not play such a role. The hypothetical “may” mode is present is many considerations, and the authors do well to suggest the need for more certainty.
The authors could additionally consider the importance of this issue specifically for the older subjects that often exhibit idiosyncratic responses that are hardly studied, as most studies are done in younger subjects, in both human and animal studies, and consider the causes for this exclusion of older subjects that limits the evidence base. Though this suggestion is not mandatory.
Over all, this review is of great interest, it is thought provoking and can stimulate discussion on this very timely and significant topic. Hence the article is recommended for publication.
Author Response
We would like to thank the reviewer for his assessment and advice.
We have added a respective sentence in the conclusion section accordingly.
Reviewer 2 Report
This is an interesting study that adds to the body of work focused on Immunosenescence in neurological diseases. The strategy used by the authors to passe through all main papers that discussed the topics is non classical using Pubmed website but gives non-exhaustive results in the studies that investigated cellular changes in the ageing immune system and in neurological disease. The authors focused on multiples sclerosis (MS), Parkinson’s disease, dementia, and stroke diseases.
The review is clearly written and well organized the introduction is reasonable and the tables are comprehensive and helpful to summarize all the studies for each disease.
It would be interesting if the authors could discuss any ideas about the Immunosenescence of stem cells in neurological diseases. As we know that there is also a dramatic drop in the neural stem cells number and function with aging (Georgios Kalamakis et al. Cell. 2019) and in aged animals, exposure to young blood through heterochronic parabiosis improves stem cell function in spinal cord and brain, and counteracts ageing and rejuvenates cognitive processes (Villeda, S. A. et al. Nat. Med. 20, 659–663 (2014)). It could be interesting if the authors address this question or add this point at the discussion. This can help to improve and complete this review.
Author Response
We would like to thank the reviewer for his assessment and suggestion. We have introduced a respective sentence in the summary section.